# Bitcoin Money Laundering Detection via Subgraph Contrastive Learning

**DOI:** 10.3390/e26030211

**Published:** 2024-02-28

**Authors:** Shiyu Ouyang, Qianlan Bai, Hui Feng, Bo Hu

**Affiliations:** 1School of Information Science and Technology, Fudan University, Shanghai 200433, China; ysyou21@m.fudan.edu.cn (S.O.); bohu@fudan.edu.cn (B.H.); 2School of Computer Science, Fudan University, Shanghai 200433, China; 17210720025@fudan.edu.cn

**Keywords:** Bitcoin, graph neural network, anti-money laundering, contrastive learning, heterogeneous graph

## Abstract

The rapid development of cryptocurrencies has led to an increasing severity of money laundering activities. In recent years, leveraging graph neural networks for cryptocurrency fraud detection has yielded promising results. However, many existing methods predominantly focus on node classification, i.e., detecting individual illicit transactions, rather than uncovering behavioral pattern differences among money laundering groups. In this paper, we tackle the challenges presented by the organized, heterogeneous, and noisy nature of Bitcoin money laundering. We propose a novel subgraph-based contrastive learning algorithm for heterogeneous graphs, named Bit-CHetG, to perform money laundering group detection. Specifically, we employ predefined metapaths to construct the homogeneous subgraphs of wallet addresses and transaction records from the address–transaction heterogeneous graph, enhancing our ability to capture heterogeneity. Subsequently, we utilize graph neural networks to separately extract the topological embedding representations of transaction subgraphs and associated address representations of transaction nodes. Lastly, supervised contrastive learning is introduced to reduce the effect of noise, which pulls together the transaction subgraphs with the same class while pushing apart the subgraphs with different classes. By conducting experiments on two real-world datasets with homogeneous and heterogeneous graphs, the Micro F1 Score of our proposed Bit-CHetG is improved by at least 5% compared to others.

## 1. Introduction

As an emerging distributed ledger technology, blockchain has been abused by a great deal of illicit activity due to its decentralized, pseudonymous, and convenience [1]. According to the report by Chainalysis [2], cybercriminals’ money laundering activities through cryptocurrencies reached USD 23.8 billion in 2022, a 68.0% increase compared to 2021. Money laundering involves transferring illicit funds through licit means and often includes criminal activities such as gambling, fraud, and human trafficking [3]. Considering the adverse impact of money laundering on society and the economy, international organizations and governments are closely monitoring the issue of cryptocurrency money laundering, aiming to strengthen regulations to curb money laundering crimes. For instance, the Financial Action Task Force updated its report on the implementation of standards for virtual assets and virtual asset service providers in 2021, urging countries to enhance the regulation of virtual assets [4]. Additionally, in 2022, the European Union passed the “Markets in Crypto-assets” cryptocurrency regulatory protocol, standardizing participants in the crypto market [5]. However, as blockchain acts as the “bank” for cryptocurrencies, with its peer-to-peer and decentralized features [6], it presents various challenges for regulation. Unlike transactions in regular banks, transactions in Bitcoin occur between addresses, which are digitally signed and verified in a public ledger on the blockchain without any intermediaries. The user is hidden behind a pseudonym in Bitcoin, rendering centralized regulatory approaches ineffective, which rely on rigorous compliance investigations and user monitoring [7]. In view of this, it is imperative to study the detection of money laundering in cryptocurrency.

Cryptocurrency anti-money laundering has garnered widespread attention in the academic community, with methods primarily falling into three categories: rule-based methods, unsupervised anomaly detection methods, and supervised machine learning methods. Rule-based algorithms typically detect illicit activities by constructing expert systems [8] or using heuristic algorithms [9,10,11,12], but these algorithms are limited by the pseudonymous and ever-changing rules of cryptocurrencies. Unsupervised AML methods achieve detection through clustering, such as trimmed *k*-means [13] and community clustering [14], classifying transactions with similar patterns as a group to help detect anomalous transactions. However, research indicates that the effectiveness of those algorithms is not comparable with supervised AML algorithms. Therefore, more and more researchers are focusing on supervised learning methods to address these challenges, using training data with known labels to train models that learn the features of normal and suspicious transactions to help regulatory authorities detect potential money laundering activities. Among these, graph-related algorithms perform exceptionally well, including a node2vec-based classifier [15], graph convolutional neural networks (GCNs) [16], and their variations [17,18,19]. Since the release of the largest supervised Bitcoin dataset by Elliptic [20], which represents transactions as nodes and flows between transactions as edges, the detection of illicit money laundering can be viewed as a node classification task.

However, detecting cryptocurrency money laundering using graph structures is highly challenging due to the organized, heterogeneous, and noisy nature of the illicit behavior.

**Organized**. Money laundering is usually an organized behavior while current algorithms in Bitcoin mainly focus on node-level detection. Therefore, it is a challenge to come up with an algorithm that detects money laundering groups directly. A large number of disclosed large-scale cryptocurrency money laundering cases (e.g., 1MDB [21] and Danske Bank scandal [22]) show that money laundering activities typically exhibit scale and organizational characteristics. As shown in Figure 1, node-level detection methods can identify individual nodes as potential illicit transactions. However, they ignore the relationships and interactions between nodes. In contrast, subgraph-level detection methods consider the topology between nodes and attempt to identify subgraphs with similar transaction patterns, such as frequent fund transfers and lengthy transaction chains.**Heterogeneous**. Although GNN-based illicit transaction detection techniques have achieved significant success, most of them are focused on homogeneous graphs, i.e., transaction record graph [20] or wallet address graph [23] in the upper-left corner of Figure 2. In reality, heterogeneity is an inherent characteristic of cryptocurrency transaction networks [24]. Specifically, the wallet address and transaction records together form the graph, as depicted in the top-left corner of Figure 2. Heterogeneity increases the complexity of data mining, leading to a more intricate risk identification process.**Noisy**. Despite the significant differences in behavioral patterns between licit and illicit transactions, real-world transactions often exhibit a notable amount of noise, including erroneous transactions and intentionally disruptive transactions initiated by money launderers to obfuscate their activities [25]. As a result, these noises can lead to an unclear distinction in the transaction topology.

**Figure 1 entropy-26-00211-f001:**
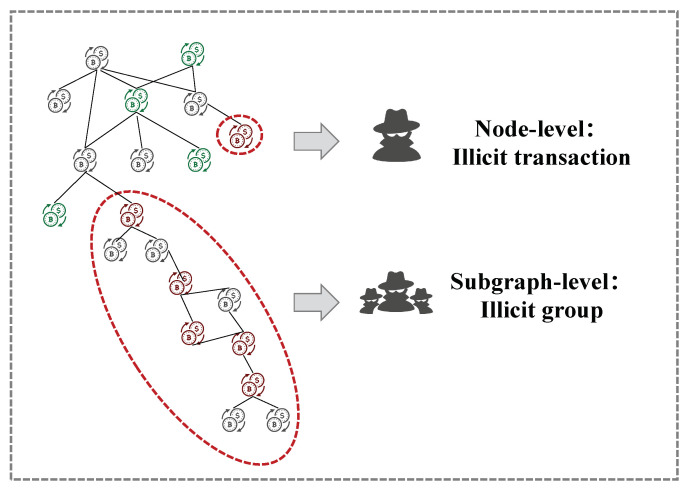
Organized behavior of an illicit group. The node-level detection methods identify the individual illicit transaction, while the subgraph-level detection methods identify the illicit group.

**Figure 2 entropy-26-00211-f002:**
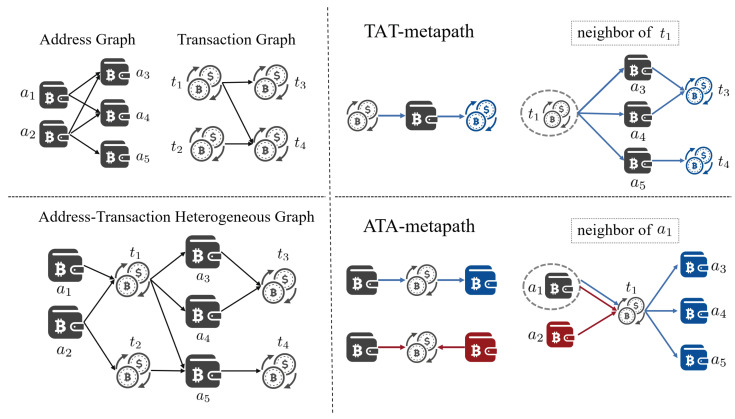
Address–transaction heterogeneous graph. The (**top-left**) is the wallet address graph and transaction record graph. The (**bottom-left**) is the address–transaction heterogeneous graph. The (**top-right**) is the neighbor of $t_{1}$ under the TAT-metapath, and the (**bottom-right**) is the neighbor of $a_{1}$ under ATA-metapath, where the dashed round box indicates the target node, the blue and red lines distinguish different path directions, and the blue and red nodes are the neighbors of the target node under the corresponding path.

Considering the above characteristics of cryptocurrency money-laundering behavior, we design a subgraph-level graph contrastive learning algorithm based on the heterogeneous information of the Bitcoin network, namely contrastive heterogeneous graph neural network (Bit-CHetG). The Bit-CHetG consists of four main components: transaction subgraph embedding (TSE), address feature aggregation (AFA), feature fusion (FF), and contrastive learning (CL). Firstly, for the heterogeneity of the network, we propose an address–transaction heterogeneous graph, as shown in the bottom-left corner of Figure 2, to establish the connection between consecutive transaction records and related wallet addresses. Thus, the TSE component and AFA component extract features under different metapaths and merge them through the FF component. Secondly, for the organizational characteristics of money-laundering behavior, the TSE component constructs downstream transaction subgraphs, and the AFA component constructs the associated address subgraphs. Specifically, we recommend the tree-structure to be the representative structure of the transaction subgraph since the flow of money laundering funds tends to be dispersed from upstream to downstream [26], which is confirmed to be effective in the experiments of Section 5.2. Thirdly, for the noise during trading, we employ graph data augmentation strategies, such as edge perturbation and node reconnection, to simulate the scenarios with inherent noise, so as to provide the model with input data filled with rich noise features. Subsequently, we introduce supervised graph contrastive learning [27] to explicitly identify differences in the augmented data and obtain a robust representation.

The contributions of this paper can be summarized as follows.

This work focuses on mining transaction patterns in subgraphs. We have discovered that the tree structure, as typical transaction patterns, can serve as a representative structure for distinguishing money laundering from non-money laundering activities.To the best of our knowledge, we are the first to propose a subgraph detection model based on graph contrastive learning methods in the field of cryptocurrency money laundering detection.Experimental results demonstrate the effectiveness of the Bit-CHetG models by integrating various money laundering detection models such as random forest [28], GCN [16], inspection-L [29], SubGNN [30], Tsgn [31], HAN [32], and MAGNN [33]. The comparison algorithms cover the latest graph-based money laundering detection algorithms in Bitcoin, subgraph classification algorithms, and heterogeneous graph classification algorithms. In particular, the Micro F1 Score of our proposed Bit_CHetG is improved by at least 5%.

The rest of the paper is organized as follows. Section 2 presents related work in cryptocurrency about money laundering detection and subgraph representation algorithms. Section 3 introduces the Bitcoin address–transaction heterogeneous graph, defining two types of metapaths, and clarifies the subgraph classification problem. In Section 4, the four modules of the proposed algorithm are presented in detail. Section 5 describes the data sources and the experimental results. In Section 6, we discuss the social and economic implications of the algorithms proposed in the paper. In Section 7, the paper is summarized, and we analyze the limitations of this research and propose future research directions.

## 2. Related Works

In this section, we will provide the necessary background knowledge for cryptocurrency about the classification of money laundering detection algorithms and subgraph representation algorithms.

### 2.1. Money Laundering Detection in Cryptocurrency

Cryptocurrency AML methods fall into three main categories: rule-based methods, unsupervised anomaly detection methods, and supervised machine learning methods.

In practice, AML for cryptocurrencies often relies on rule-based algorithms [34], such as building expert systems using domain knowledge to detect money laundering activities [8], or employing heuristic algorithms for address identity inference [9,10,11,12] to enhance the transaction traceability and further uncover illicit transactions. For example, the process of money laundering involves complicated financial activities that may exhibit distinctions from normal ones, such as large-block or high-frequency transactions, the reactivation of dormant addresses, and the immediate closures of newly opened addresses [35]. Zhou et al. [36] defined a feature system consisting of 40 statistical features to characterize a money laundering transaction behavior based on that domain knowledge. However, most rule-based algorithms are based on empirical inferences and may become ineffective when facing rule changes.

Currently, the label-free anomaly detection method has become an important method for AML in Bitcoin. Most unsupervised illicit behavior detection methods [37,38,39] aimed to find behavioral patterns that differ significantly between illicit cases (the minority) and licit cases (the majority). Among them, clustering is the most common approach. Monamo et al. [13] studied the use of trimmed *k*-means to detect fraudulent activity in Bitcoin transactions. Various graph centrality measures (i.e., in degree, out-degree of the Bitcoin transactions) and currency features (i.e., the total amount sent) were used for Bitcoin transaction clustering. The modified *k*-means clustering methods, such as local outlier factor (LOF), are adopted to detect suspicious behavior on two graphs generated by Bitcoin transactions [39], one graph with users as nodes and the other with transactions as nodes. The two types of graphs consider the unspent transaction output (UTXO) model of Bitcoin, where each transaction output is associated with a certain number of cryptocurrencies initiated by a specific user. However, the construction of these graphs does not take into account the rich information of heterogeneous networks, which will be emphasized by our proposed algorithm.

Furthermore, entropy can also help identify unusual patterns or behaviors in the Bitcoin network [40], providing a possible perspective for preventing cryptocurrency money laundering. As a concept in information theory, entropy [41] denotes the degree of uncertainty or chaos in a system. Liu et al. [42] used different entropies to describe the degree of chaos in the cryptocurrency market, such as crypto-economic entropy and Kolmogorov entropy. Recently, entropy is also used for anomaly detection in cryptocurrency networks [43], which includes malicious actions, attacks, and illicit behaviors such as money laundering. For example, one pioneering work by Pham et al. [39] combined entropy with clustering algorithms to optimize the clustering effect of cryptocurrency transaction graphs using cross-cluster entropy [44] in order to improve the accuracy of identifying illicit transactions. They measure the quality of clusters by calculating the entropy of the data distribution between different clusters to find the optimal number of clusters.

Nevertheless, extensive experimental results have shown that using unsupervised anomaly detection methods is insufficient for detecting illicit patterns in a real-world Bitcoin transaction dataset [45]. Therefore, our paper mainly focuses on supervised learning algorithms for Bitcoin AML. Yining et al. [15] utilized “Wallet Explorer” to collect Bitcoin transaction data and found that the classifier based on node2vec [46] outperformed the clustering methods in detecting money laundering transactions. However, their approach solely focuses on graph topological patterns and does not take the node features into account. In recent years, Elliptic [47], a cryptocurrency intelligence company dedicated to protecting cryptocurrency systems, has launched the Elliptic dataset, which contains local features directly related to specific transactions and aggregated features from neighboring transactions. This dataset has facilitated research in the AML community and the development of machine learning techniques. In addition, the dataset labels nodes as licit and illicit, further defining the problem of money laundering detection as a node classification problem on the graph. Previous researchers have explored various approaches for utilizing the Elliptic dataset, including classical supervised machine learning methods [48,49], EvolveGCN for dynamic graphs [50], signature vectors in blockchain transactions (SigTran) model [51], and so on. Moreover, Weng et al. [29] were the first to apply self-supervised GNNs to research the Bitcoin AML problem. They proposed a graph neural network framework called Inspection-L, which exploited unknown labeled data in a self-supervised manner, thus improving the quality of representation for downstream tasks such as Bitcoin money laundering detection. In contrast to self-supervised contrastive learning strategies, our approach makes full use of label information, preventing instances of the same label such as the anchor being mixed into negative samples. In addition, since criminals can intentionally mimic normal behaviors, we employ data augmentation techniques to introduce noise and simulate real-world scenarios.

It is worth noting that the works mentioned above treated money laundering detection as a node classification problem rather than subgraph classification, resulting in the neglect of local topology.

### 2.2. Subgraph-Based Representation in Cryptocurrency

More and more subgraph mining methods such as complex network analysis as well as motif matching methods are applied to cryptocurrency anomalous transaction mining [52].

Some researchers [23,53,54] have used relevant the metrics of complex networks to study the anomalous subgraphs in Bitcoin transaction graphs. Tao et al. [54] sampled subgraphs by a random walk with flying-back properties to observe the network’s small-world phenomenon, polycentric state, etc., which provides insights for malicious activities and fraud detection in cryptocurrency blockchain networks. Xiang et al. [23] conducted a deep dive into the transaction patterns of the Bitcoin address network through complex networks and machine learning, and they found that, in a suspected money laundering subgraph, the original address would simultaneously send Bitcoins to both illicit addresses and regular addresses through one-many transactions. Money laundering groups employ multi-round top–down transactions to evade the tracking of the original Bitcoin, and this pattern inspired our subgraph mining algorithm. Some scholars have studied motifs on blockchain transaction networks for price prediction [55], network attribute analysis [56], and exchange pattern mining [12]. Motif is defined as the cyclic subgraph patterns of the networks, Wu et al. [57] built a feature-based network analysis framework based on hybrid motifs to identify the statistical attributes of money laundering and mixing services from three levels, namely the network level, account level, and transaction level. Considering that the complexity of matching motifs increases exponentially with the growth in nodes, the mentioned papers typically use size-2 and size-3 motifs as basic structures. However, money laundering groups often have long chains, making it difficult for motif-based methods to mine the key patterns. The subgraph mapping network proposed by Tsgn [31] has achieved good results in Ethernet anomaly pattern mining; however, only 1-hop neighborhood information is collected, resulting in limited effectiveness for money laundering pattern mining. In contrast, our Bit-CHetG can learn representations for larger subgraphs in the Bitcoin network.

In recent years, several neural network-related algorithms performed well in subgraph pattern mining, e.g., Cluster-GCN [58], SubGNN [30], and GCC [59] use subgraphs to design more efficient and scalable algorithms for training deep and large-scale GNNs and predicting subgraphs. However, employing neural networks to identify suspicious money laundering groups in large cryptocurrency transaction networks remains an open challenge. In this paper, we construct an address–transaction heterogeneous graph and aim to find representative subgraph structures to distinguish money laundering patterns in Bitcoin.

## 3. Problem

Our research is dedicated to the detection of money laundering groups in cryptocurrency. Therefore, in this section, we formulate the Bitcoin address–transaction heterogeneous graph based on the smallest transaction unit of Bitcoin, including the input and output. Subsequently, we define two types of metapaths on the heterogeneous graph, which serve as the foundation to sample the subgraphs. Finally, we define the illicit subgraph detection problem. The symbols are defined in Table 1.

**Definition 1.** 
*Address–transaction heterogeneous graph. The heterogeneous graph is denoted by $G_{\mathrm{het}}=\left\{T,A,E_{T\to A},E_{A\to T}\right\}$, including the node sets and edge sets. The address–transaction heterogeneous graph is illustrated in the bottom-left corner of Figure 2. The transaction node-set and address node-set are denoted as T and A, respectively. Each t∈T is a transaction record that consists of a set of attributes, such as in-amount, out-amount, fee, and so on. Similarly, there are some features associated with a wallet address a∈A such as the total amount, number of transactions, and so on. Then, the input feature matrix is denoted by $X_{t}\in R^{\left|T\right|\times d_{1}}$ and $X_{a}\in R^{\left|A\right|\times d_{2}}$, where $d_{1}$ and $d_{2}$ are the dimensions of the feature vector, |T| is the total number of transactions, and |A| is the total number of addresses. Moreover, there exist two types of edges from the source node to the target node, $E_{A\to T}\in E_{A\to T}$ represents an edge where a wallet address initiates a transaction and $E_{T\to A}\in E_{T\to A}$ represents an edge where a transaction outputs a certain wallet address. Note that the subscript A indicates that the node type is an address and T indicates the transaction node type. Therefore, the address–transaction heterogeneous graph not only contains the initiating and receiving addresses of a particular transaction but also demonstrates the connections between transactions and addresses during successive trading.*


**Definition 2.** 
*TAT-metapath. The transaction–address–transaction metapath, denoted by $\Phi_{\mathrm{TAT}}$, consists of one type of metapath $T\overset{E_{T\to A}}{⟶}A\overset{E_{A\to T}}{⟶}T$, which is indicated by the blue line in the top-right corner of Figure 2.*


The TAT-metapath represents a way where two transaction records are connected through the same wallet address. The neighborhood set, denoted as $N_{t}^{\Phi_{\mathrm{TAT}}}$, contains the homogeneous neighborhood adjacent to the transaction node *t* via the TAT-metapath. Therefore, the transaction neighborhood contains the set of downstream transactions associated with transaction *t*. In the top-right corner of Figure 2, the node set $\left\{t_{3},t_{4}\right\}$, which is pictured in blue, includes the downstream adjacent transactions of the transaction node $t_{1}$ based on the 1-hop TAT-metapath. Thus, we can sample the transaction subgraph connected by TAT-metapath, with the set denoted as $G_{\mathrm{sub}}^{T}=\left\{G_{1}^{T},\ldots ,G_{m}^{T},\ldots ,G_{M}^{T}\right\}$, where *m* is the index of the transaction subgraph and *M* is the total number of transaction subgraphs. There, $G_{m}^{T}=\left\{V_{m}^{T},E_{m}^{T}\right\}$, which is a connected subgraph, represents the *m*-th transaction subgraph.

It is important to note the edge direction when defining the TAT-metapath. When detecting illicit transactions, we continuously monitor the downstream transactions related to suspicious transactions through $T\overset{E_{T\to A}}{⟶}A\overset{E_{A\to T}}{⟶}T$, thus disregarding the reverse metapath. Meanwhile, we exclude the same-level neighbors under $T\overset{E_{T\to A}}{⟶}A\overset{E_{T\to A}}{⟵}T$ because the downstream neighbors under $T\overset{E_{T\to A}}{⟶}A\overset{E_{A\to T}}{⟶}T$ will cover the related nodes. As shown in Figure 3, where $T\overset{E_{T\to A}}{⟶}A\overset{E_{T\to A}}{⟵}T$ is indicated by the red dotted line in Figure 3a, and $t_{2}$ is considered as the same-level neighbor of $t_{1}$ under that path in Figure 3b, then both $t_{1}$ and $t_{2}$ will also be included in the downstream neighbors of $t_{0}$ in Figure 3c (perhaps after multiple hops). In this case, even if both $t_{1}$ and $t_{2}$ are illicit, excluding $T\overset{E_{T\to A}}{⟶}A\overset{E_{T\to A}}{⟵}T$ does not lead to the disappearance of $t_{2}$, since they will appear in the transaction subgraph of the source illicit node. The experimental results about sampling by polytree in Section 5.3 also demonstrate the rationality of that design.

**Definition 3.** 
*ATA-metapath. The address–transaction–address metapath, denoted by $\Phi_{\mathrm{ATA}}$ in the bottom-right corner of Figure 2, is defined as a combination of two types of metapaths $A\overset{E_{A\to T}}{⟶}T\overset{E_{T\to A}}{⟶}A$ in blue line and $A\overset{E_{A\to T}}{⟶}T\overset{E_{A\to T}}{⟵}A$ in red line.*


The ATA-metapath represents a way for wallet addresses to be connected by jointly participating in a transaction. The neighborhood set, denoted by $N_{a}^{\Phi_{\mathrm{ATA}}}$, contains the homogeneous neighborhood adjacent to the address node *a* via the ATA-metapath. Therefore, the address neighborhood contains the set of initiating wallet addresses and receiving wallet addresses associated with the target transaction *t*. In the bottom-right corner of Figure 2, for example, address $a_{1}$ has 1-hop neighborhood set $\left\{a_{3},a_{4},a_{5}\right\}$, which is pictured in blue, based on the metapath $A\overset{E_{A\to T}}{⟶}T\overset{E_{T\to A}}{⟶}A$ and $\left\{a_{2}\right\}$, which is pictured in red, based on $A\overset{E_{A\to T}}{⟶}T\overset{E_{A\to T}}{⟵}A$, which represents the associated wallet addresses for transaction $t_{1}$. Thus, based on the ATA-metapath, we sample the addresses associated with the transaction node *t* into an associated address subgraph, denoted by $G_{t}^{A}$. And, the set of address subgraphs is denoted by $G_{\mathrm{sub}}^{A}=\left\{G_{1}^{A},\ldots ,G_{t}^{A},\ldots ,G_{\left|T\right|}^{A}\right\}$, where the index represents the target transaction node *t*. Note that the total number of associated address subgraphs is |T| because each transaction node can generate an associated address subgraph.

Our objective is to find a suitable sampling method to obtain the subgraphs $G_{\mathrm{sub}}^{T}$ and $G_{\mathrm{sub}}^{A}$ from the global heterogeneous graph $G_{\mathrm{het}}$ that facilitates distinguishing the patterns of money laundering and non-money laundering, and then learn a function $f:G_{\mathrm{sub}}^{T}\times G_{\mathrm{sub}}^{A}\mapsto R$ to predict the probability that each subgraph is illicit. Given the transaction subgraph $G_{\mathrm{sub}}^{T}=\left\{G_{1}^{T},\ldots ,G_{m}^{T},\ldots ,G_{M}^{T}\right\}$, each transaction subgraph $G_{m}^{T}$ is associated with a label $c_{m}$, indicating the percentage of illicit nodes in each subgraph. Note that, while this paper focuses on the transaction subgraph classification task, the methods we propose entail learning a subgraph classifier $f:G_{\mathrm{sub}}^{T}\mapsto \left\{1/N,2/N\ldots N/N\right\}$, while *N* is the size of the subgraph.

## 4. Proposed Method

In the following section, we will provide a detailed explanation of our proposed model for Bitcoin money laundering group detection, named Bit-CHetG. As shown in Figure 4, the model comprises four main components: transaction subgraph embedding, address feature aggregation, feature fusion, and contrastive learning. Leveraging the predefined address–transaction heterogeneous graph and metapaths, we first extract topology features for the transaction node and aggregated features for the address node from different metapaths. Specifically, for the TAT-metapath $\Phi_{\mathrm{TAT}}$, the TSE component conducts multi-step sampling on the transaction neighborhood set $N_{t}^{\Phi_{\mathrm{TAT}}}$ to obtain tree-structured transaction subgraphs and then utilizes graph neural networks to derive the topological embedding representation. On the other hand, for the ATA-metapath $\Phi_{\mathrm{ATA}}$, the AFA component, which obtains the associated address representation, employs the graph neural networks to embed the associated address subgraphs of the target transaction node based on the neighbors $N_{a}^{\Phi_{\mathrm{ATA}}}$. Subsequently, the FF component aggregates these associated address representations into the topological embedding representation and obtains the fused features. Finally, to enhance the classification accuracy of the model, we introduce the CL component, a supervised contrastive learning approach, to learn and classify the fused features of transaction subgraphs from both the same and different classes.

### 4.1. Transaction Subgraph Embedding

In the Bitcoin network, there are evident topological differences between money laundering transaction subgraphs and non-money laundering transaction subgraphs, as extensively described in Figure 5. To address this, we focus on the TAT-metapath in the Bitcoin address–transaction heterogeneous graph. Thus, we perform transaction subgraph sampling and then utilize GCN [16] to obtain the topological embedding matrix of the transaction subgraph, denoted by $H^{\Phi_{\mathrm{TAT}}}$.

#### 4.1.1. Transaction Subgraph Sampling

In order to summarize the structural commonalities of transaction subgraphs with different sizes, we adopt an *n*-hop sampling method, where *n* is the number of hops, to generate the licit and illicit subgraphs, respectively.

When generating the illicit subgraph, we start with an illicit transaction node and expand it by *n* hops. If all terminal nodes are licit, the process stops; otherwise, continue expanding by *n* hops.When generating a licit subgraph, we initiate the process from a licit transaction node and stop the extension after *n* hops. However, if the generated subgraph contains any illicit nodes, it is considered to be illicit. This condition ensures that the generated licit subgraph maintains its legality.

The above *n*-hop sampling method yields a tree structure that is similar to the flow of money during cryptocurrency trading. And, the typical topology of the subgraphs is displayed in Figure 6. The relevant conclusions show that the sampling method with a tree structure helps distinguish the transaction patterns of money laundering groups.

For the subgraph sampling process of the Bit-CHetG algorithm, we fix the size of each transaction subgraph as *N*, while the optimal value of *N* is determined through parametric experiments in Section 5.4. In Bit-CHetG, we adopt a breadth-first random walk approach to sample transaction nodes from the address–transaction heterogeneous graph, generating a series of transaction subgraphs $G_{m}^{T}$. The generated subgraph only contains transaction nodes and is tree-structured.

**Step 1:** Given a target transaction node *t* as the parent node *p*, add the 1-hop neighborhood set $N_{t}^{\Phi_{\mathrm{TAT}}}$ of node *t* based on the TAT-metapath into the node set $V_{m}^{T}$ of the subgraph $G_{m}^{T}$. The edges between node *t* and the nodes in the neighborhood set $N_{t}^{\Phi_{\mathrm{TAT}}}$ are added to the edge set $E_{m}^{T}$ of the subgraph. As shown by the subgraph sampling process framed by the dashed line in the bottom-left corner of Figure 4, the first generated subgraph contains the parent node framed in blue, as well as 1-hop neighbor nodes.**Step 2**: Randomly select a node from the 1-hop neighborhood set $N_{t}^{\Phi_{\mathrm{TAT}}}$ with a certain probability and extend the subgraph according to Step 1 with this node as the new parent node. This process generates the second subgraph and third subgraph shown in the subgraph sampling dashed line of Figure 4. If all the neighbor nodes have been traversed, continue to extend to the next level until the number of nodes in the node-set $V_{m}^{T}$ reaches a fixed number *N*.

During the subgraph sampling process, we use the breadth-first algorithm to sample downstream nodes from the parent node to ensure that the transaction subgraphs sampled by the TAT-metapath exhibit a tree-like structure. Through this sampling process, we can generate highly correlated subgraphs to better capture the relationships between transaction nodes. Figure 4 in the bottom-left corner shows the sampling process of a transaction subgraph.

#### 4.1.2. Topology Feature Embedding

Based on the transaction subgraph $G_{m}^{T}=\left\{V_{m}^{T},E_{m}^{T}\right\}$, which has an adjacency matrix $A_{m}\in R^{N\times N}$ and a feature matrix $X_{m}\in R^{N\times d_{1}}$, we proceed with GCN to extract high-order subgraph representation. The forward propagation process is as follows: (1)$$H^{\left(l\right)}=\sigma \left({\hat{A}}_{m}H^{\left(l-1\right)}W^{\left(l-1\right)}\right),$$
where σ is the activation function, ${\hat{A}}_{m}={\tilde{D}}^{-\frac{1}{2}}\tilde{A}{\tilde{D}}^{-\frac{1}{2}}$ is the normalized adjacency matrix, $\tilde{A}=A+I$ is the adjacency matrix with added self-connections, $D_{ii}=\sum_{j}{\tilde{A}}_{ij}$ represents the degree of the *i*-th node, $W^{\left(l-1\right)}\in R^{d_{l-1}\times d_{l}}$ is the weight matrix of the (*l* − 1)-th layer, $H^{\left(l-1\right)}\in R^{N\times d_{l-1}}$ is the hidden layer representation matrix of the (*l* − 1)-th layer, and the initial feature matrix $H^{\left(0\right)}=X$.

Finally, we consider the feature matrix of the last layer as the final graph embedding representation of the transaction subgraph: (2)$$H^{\Phi_{\mathrm{TAT}}}=H^{\left(L\right)},$$
where $H^{\Phi_{\mathrm{TAT}}}\in R^{N\times d_{T}}$, *L* is the number of layers of the graph neural network, and $d_{T}$ is the feature dimension of the last layer.

Note that $h_{t}^{\Phi_{\mathrm{TAT}}}\in R^{d_{T}}$ is the topological embedding the representation of the transaction node *t*, which is the node feature vector obtained from the final layer of the graph neural network.

### 4.2. Address Feature Aggregation

In the Bitcoin network, each transaction may involve multiple input and output addresses. To capture the relationships between the target transaction and its associated addresses, we focus on the ATA-metapath predefined in the Bitcoin heterogeneous graph. The purpose of this section is to learn the associated address representation, denoted by $h_{t}^{\Phi_{\mathrm{ATA}}}$, of the target transaction nodes.

As illustrated by the blue box in the address–transaction heterogeneous graph in Figure 4, one wallet may connect to multiple transactions. Thus, the set of one-hop neighbors generated through the ATA-metapath, denoted by $N_{a}^{\Phi_{\mathrm{ATA}}}$, may include wallet addresses associated with multiple transaction records. When constructing the associated address subgraph $G_{t}^{A}$, we are only interested in the neighbor set relevant to the target transaction node *t*, which is a subset of $N_{a}^{\Phi_{\mathrm{ATA}}}$. The process of constructing the associated transaction subgraph can be simplified as follows:**Step1: Identify the target node.** Firstly, designate $t_{1}$ as the target transaction node for the AFA module, and add $t_{1}$ to the node set $N_{t}^{A}$, highlighted by the red dashed box in Figure 4.**Step2: Determine the edge set and node set.** Traverse the node and edge sets in the address–transaction heterogeneous graph. Add all one-hop neighbor address nodes of the target node $t_{1}$ to the node set $N_{t}^{A}$. These address nodes are connected through the ATA-metapath. The edge set $E_{t}^{A}$ consists of edges connecting the target node $t_{1}$ and the address nodes, without distinguishing the direction of edges.

After obtaining the associated address subgraph $G_{t}^{A}$, we use a GCN model similar to Section 4.1.2 for feature mapping. The layer update formula for heterogeneous graph convolutional networks can be expressed as follows, where $H_{i}^{\left(l+1\right)}$ is the node representation matrix of node type *i* in layer l+1:(3)$$H_{i}^{\left(l+1\right)}=\sigma \left(\sum_{j\in N_{i}}D_{i}^{-\frac{1}{2}}A_{ij}D_{j}^{-\frac{1}{2}}H_{j}^{\left(l\right)}W_{ji}^{\left(l\right)}+X_{i}W_{ii}^{\left(l\right)}\right),$$
where $H_{i}^{\left(l\right)}$ is the node representation matrix of node type *i* in layer *l*, $N_{i}$ is the set of neighbor node types of node type *i*, $A_{ij}$ is the adjacency matrix from node type *j* to node type *i*, $D_{i}$ is the degree matrix of node type *i*, $W_{ji}^{\left(l\right)}$ and $W_{ii}^{\left(l\right)}$ are learnable weight matrices for information transfer from node type *j* to node type *i*, $X_{i}$ is the feature matrix of node type *i*, and σ is the activation function.

Thus, the embedding matrix of the associated address subgraph, denoted by $H^{\Phi_{\mathrm{ATA}}}$, is represented by the feature matrix of the last layer:(4)$$H^{\Phi_{\mathrm{ATA}}}=H^{\left(L\right)},$$
where *L* is the number of layers of the graph neural network.

Since our goal is to obtain the associated address representation of the target transaction node *t*, denoted by $h_{t}^{\Phi_{\mathrm{ATA}}}$ in Figure 4, we use average pooling [60] as the readout function to generate the subgraph-level representations. It is formulated as follows: (5)$$h_{t}^{\Phi_{\mathrm{ATA}}}=\frac{1}{K}\sum_{j=1}^{K}H^{\Phi_{\mathrm{ATA}}}\left[j,:\right],$$
where *K* is the size of the address subgraph, $h_{t}^{\Phi_{\mathrm{ATA}}}\in R^{d_{A}}$, and $d_{A}$ is the dimension of the associated address representation.

### 4.3. Feature Fusion

For each transaction subgraph $G_{m}^{T}$, we obtained the topological embedding representation $h_{t}^{\Phi_{\mathrm{TAT}}}$ of the individual transaction node *t* encoded by GCN, as well as the associated addresses representation $h_{t}^{\Phi_{\mathrm{ATA}}}$ for the target transaction node *t*. We then concatenate these two representations to obtain a fused feature representation for the individual transaction node *t*: (6)$$h_{t}=\mathrm{Concat}\left(h_{t}^{\Phi_{\mathrm{TAT}}},Wh_{t}^{\Phi_{\mathrm{ATA}}}\right),$$
where $W\in R^{d_{T}\times d_{A}}$ is the weight matrix, to align the two represented dimensions.

Next, we use a readout function such as average pooling to obtain the graph-level fused feature vector for each transaction subgraph, denoted by $g_{m}$, where the subscript *m* indicates the *m*-th subgraph: (7)$$g_{m}=\mathrm{ReadOut}\left(h_{t}|\forall t_{i}\in G_{m}^{T}\right).$$

Finally, we use a fully connected (FC) layer followed by a softmax layer to obtain the predicted scores for the illicit rate of each transaction subgraph: (8)$${\hat{c}}_{m}=\mathrm{Softmax}\left(\mathrm{FC}\left(g_{m}\right)\right).$$

For the *m*-th transaction subgraph $G_{m}^{T},c_{m}$ represents the ground truth label for its illicit rate. The prediction loss function is then defined as:(9)$$L_{\mathrm{MEA}}=\sum_{m=1}^{M}\left|c_{m}-{\hat{c}}_{m}\right|.$$

### 4.4. Contrastive Learning

In the last section, we introduce contrastive learning and construct a new loss function as the optimization objective for illicit transaction subgraph detection. The goal is to maximize the consistency among positive samples while minimizing the similarity with negative samples, given the provided labels. Specifically, for a given subgraph $G_{a}^{T}\in G_{\mathrm{sub}}^{T}$ considered as an anchor, we treat subgraphs with the same label as positive trees $G_{p}^{T}$, and those with different labels as negative trees $G_{n}^{T}$. Let $G_{a}^{T}$ be an illicit transaction subgraph, and we aim to increase the similarity between its feature representation vector and those of other illicit transaction subgraphs in high-dimensional space while decreasing the similarity with the feature representations of licit transaction subgraphs. This helps the model learn time-invariant representations. This is a form of supervised contrastive learning, where we bring similar instances closer and push dissimilar instances apart, as shown in Figure 7. The final loss function consists of two components, the prediction loss, and the contrastive loss:(10)$$L=L_{\mathrm{MEA}}+\alpha L_{\mathrm{CL}},$$
where α is the hyperparameter and the contrastive loss $L_{CL}$ is calculated as follows:(11)$$L_{\mathrm{CL}}=-\sum_{m=1}^{M}\log\left\{\frac{1}{\left|P(m)\right|}\times \sum_{p\in P(m)}\frac{\exp\left(\mathrm{sim}\left(g_{m},g_{p}\right)\tau \right)}{\sum_{n\in N(m)}\exp\left(\mathrm{sim}\left(g_{m},g_{n}\right)\tau \right)}\right\}.$$

During the traversal process of the contrastive loss, the index *m* represents the anchor. Index *p* corresponds to positive samples with the same label as the anchor, which forms the positive index set $P\left(m\right)=\left\{p:c_{p}=c_{m}\right\}$. Index *n* corresponds to negative samples with different labels from the anchor *m*, which forms the negative index set $N\left(m\right)=\left\{n:c_{n}\neq c_{m}\right\}$. The function exp(·) represents the exponential function, and τ>0 is a scalar temperature parameter. The function sim(·) represents the cosine similarity function: (12)$$\mathrm{sim}\left(g_{m},g_{p}\right)=\frac{g_{m}^{T}\cdot g_{p}}{∥g_{m}^{T}∥∥g_{p}∥}.$$

In addition, data augmentation techniques are applied to simulate noise during the transaction process, such as edge perturbation and node reconnection, as shown by the dotted line of the anchor in Figure 7. For example, we introduce erroneous transactions or intentional interference transactions initiated by money launderers to hide their activities. This data augmentation process helps the model be more robust and better generalizes to real-world scenarios where noise and anomalies are present.

## 5. Experiment

In this section, to validate the effectiveness of the algorithm, we first introduce the selected dataset and comparison models. We then evaluate the proposed Bit-CHetG method by comparing it with the comparison models. The results demonstrate that there is a significant improvement in the micro F1-score by more than 5%.

Additionally, in order to explore the reasons for the effectiveness of the algorithm, we first elucidate that the choice of tree structure as a subgraph can effectively distinguish the topological patterns of cryptocurrency illicit groups. Then, we verify the enhancement of the results by the AFA module and the CL module, i.e., the introduction of contrastive learning and wallet address information.

Simultaneously, we conduct experiments to determine the optimal parameters, such as the layers of GCN and the size of the sampling subgraph.

### 5.1. Experimental Setup

#### 5.1.1. Datasets

Two datasets were selected for this paper, the Elliptic dataset [20] (a publicly available benchmark) and the BlockSec dataset (a heterogeneous dataset we collected). The statistics of the datasets are shown in Table 2. The main difference between these two datasets is that the BlockSec dataset contains transaction record information and wallet address information, while the Elliptic dataset contains only transaction record information.

**Elliptic dataset**: Provided by Elliptic [47], this is the largest labeled Bitcoin transaction dataset. The Elliptic dataset [20] comprises 203,769 transaction nodes and 234,355 transaction payment edges (i.e., transaction inputs and outputs). Within the Elliptic dataset, 21% of nodes are labeled as licit, while only 2% are marked as illicit. These nodes include 166 features, with the initial 94 features encompassing local transaction information, including *time steps, transaction fees, and input or output amounts*. The remaining 72 features are aggregated features, consisting of transaction information aggregated from neighboring nodes in a 1-hop graph, such as *standard deviation, minimum, maximum, and correlation coefficients*.

**BlockSec dataset**: Provided by BlockSec [61]. This dataset includes wallet address information and the transaction record information of Bitcoin for April 2023. The raw data include 16,674,890 transaction records involving 100,061 wallet addresses, with 1442 addresses labeled as illicit. The transaction record features include fees, input amounts, and output amounts. The wallet address features include *the total number of input and output transactions, the amount of the transactions*, and *more information related to the given address*.

Next, we conducted subgraph sampling on both datasets, using the subgraph illicit rates as the label, which represents the proportion of illicit nodes in each sampling subgraph. The processing procedure for the two datasets is as follows: Since the BlockSec dataset is a heterogeneous graph, it undergoes the transaction subgraph sampling steps in Section 4.1.1 and the associated address subgraph sampling steps in Section 4.2. In contrast, the Elliptic dataset is a homogeneous graph, and we omit the sampling of the associated address subgraph. And, the transaction subgraph sampling of the Elliptic dataset employs a similar breadth-first algorithm for downstream expansion. However, the generation of the 1-hop neighborhood set is based on $T\overset{E_{T\to T}}{⟶}T$ instead of the TAT-metapath. As shown in Table 2, the size of the transaction subgraph is fixed at 10 for the Elliptic dataset and 5 for the BlockSec dataset based on the experiments in Section 5.2 and Section 5.4. An example of the typical subgraphs of both datasets can be found in Figure 5. During the training of these two datasets, the BlockSec dataset follows the four steps in Figure 4, while the Elliptic dataset only undergoes topology feature embedding and contrastive learning.

#### 5.1.2. Comparison Algorithms

We utilize the following widely used graph-based methods, including heterogeneous and homogeneous graphs, as comparative methods for Bitcoin money laundering detection to emphasize the effectiveness of our proposed method. It is worth noting that some node-level classification algorithms are extended to become graph-level classification methods through a unified readout function [62].

For the Elliptic dataset, which only contains transaction record features and does not include wallet address features, we choose the homogeneous algorithm as the comparison method.

**Random forest** [28]: A supervised learning method used for ensemble learning enhances the predictive ability by combining results from multiple decision trees. In previous experiments focused on identifying illicit nodes in the Elliptic dataset, random forest demonstrated an outstanding performance when compared to the graph neural network algorithm.

**GCN** [16]: The fundamental graph neural network, which can extract topological features among transaction nodes. Here, we utilize GCN to obtain node representations before proceeding with subgraph classification.

**Inspection-L** [29]: The first application of a self-supervised GNN to the Bitcoin money laundering detection problem, which has a self-supervised Deep Graph Infomax framework combined with a supervised learning algorithm, random forest.

**SubGNN** [30]: It is a subgraph-based neural network that proposes three property-aware channels that capture position, neighborhood, and structural information to propagate the information at the subgraph layer.

**Tsgn** [31]: It introduces a network mapping strategy from node to edge to fully capture the potential topological information of the subgraph which cannot be easily obtained from raw transaction networks, benefiting the subsequent fraud detection algorithms in cryptocurrency.

For the BlockSec dataset, which includes both transaction record features and wallet address features, we selected the heterogeneous network algorithm for the comparative experiment.

**HAN** [32]: It is designed for heterogeneous graphs, which proposes the usage of a graph neural network with hierarchical attention to evaluating node weights and metapaths. Additionally, HAN is a semi-supervised heterogeneous method.

**MAGNN** [33]: It is a heterogeneous graph embedding model that utilizes metapath-guided aggregation to acquire meaningful node representations by considering both structural relationships and attribute information.

Our experimental environment is as follows: the operating system is Ubuntu 18.04, the programming language is Python 3.8.13, the framework is PyTorch 1.4.0, the CPU is Intel Core i7-6800K, and the GPU is GeForce GTX 1080, which is designed by NVIDIA, Santa Clara, CA, USA.

The experimental parameters are set as follows: The epoch is set to 200, iterations are set to 100, and early stopping [63] is applied when loss stops decreasing for 10 epochs. The number of layers in GCN is set to 2, and the feature dimensions of the hidden and output layers are both set to 64. The length of the metapath is set to 1 to ensure the generation of one-hop neighbors. Stochastic gradient descent is used to update the parameters of Bit-CHetG, and our model is optimized using the Adam optimizer [64]. We explore learning rates ranging from 1 × 10^−4^ to 5 × 10^−3^. For the dropout rate, we experiment with values from 0.1 to 0.5 in increments of 0.05, while hyperparameters are tuned between 0.1 and 0.9 in increments of 0.05.

Following the configuration of [20], the initial 35 graphs of the Elliptic dataset are designated as the training set, and the remaining graphs are reserved for testing. The BlockSec dataset is randomly divided into training and testing sets in a ratio of 7:3. In our dataset, the distribution of labels (i.e., subgraph illicit rates) is unbalanced, so we choose micro precision (Micro-Prec.), micro-recall (Micro-Rec.), and micro F1-score (Micro-F1) as the evaluation metrics for subgraph multiclassification problems. To ensure a fair comparison, we use the base implementation for all models and hyperparameter sweeps as in our Bit-CHetG approach. Additionally, to achieve graph-level classification, we use average pooling as a readout function [62] for methods [16,28,29,32,33]. In order to extend the homogeneous benchmarking approach to heterogeneous graphs, we fuse the original transaction features and the associated address features to obtain aggregated features, which are used as input features for the method [16,28,29,30,31].

### 5.2. Mining Tree-Structured Subgraphs

In this part, we show that the results observed from the tree-structure subgraphs and find that the topological patterns can be effectively distinguished between money laundering and non-money laundering.

According to the *n*-hop sampling method proposed in Section 4.1.1, Figure 5 illustrates the typical topology of the tree-structure subgraphs of illicit and licit transactions, respectively, where the size of the subgraph increases from left to right. Thus, we observed that:There is a significant difference between the illicit and licit subgraphs. The distribution of licit transaction trees is more centralized, similar to a network-like structure, while the distribution of illicit transaction trees is more dispersed, similar to a chain-like structure. This suggests that illicit and licit transactions exhibit different topologies and that the tree-like subgraph generation method can effectively distinguish between money laundering and non-money laundering transaction patterns.In the set of illicit subgraphs, there are continuous money laundering chains in the transaction network. Therefore, identifying individual illicit nodes can be of great help in the subsequent tracking of illicit groups.

Based on this observation, our algorithm focuses on the pattern differences between the subgraphs and chooses a tree-like structure as a typical subgraph structure.

Furthermore, we counted the changes in the size of the transaction subgraphs *N* when the number of sampling hops *n* is varied. As shown in Figure 6, the average size of transaction subgraphs in the BlockSec dataset (only transaction nodes are counted) is less than that in the Elliptic dataset for the same number of sampling hops. While the collection time is not the same for the two datasets, this phenomenon may be related to the rapid growth of mixing services [65] in recent years, and some users may use Bitcoin mixing services to enhance the privacy of their transactions. These services mix multiple transactions, making the transaction path on the chain more complex. Another possible reason is that, out of a sense of security, some users may periodically change the address they use to receive Bitcoin. This behavior can lead to truncated transaction paths in the graph because the new address is no longer associated with the previous address. This indicates that the optimal subgraph sampling size may change as the cryptocurrency ecosystem evolves. Combined with the parametric analysis of *N* performed in Section 5.4, we finally fix *n* to 5. For the following experiments in Section 5.3, *N* is fixed to 5 for the BlockSec dataset and 10 for the Elliptic dataset.

### 5.3. Experimental Results

In this part, we give THE experimental results and explain why our Bit-CHetG outperforms the comparison algorithm for four reasons.

For the elliptical dataset, we constructed 24,533 tree-structured subgraphs, starting with the labeled nodes in the original dataset. The illicit rate was then used as the label for each subgraph. For the BlockSec dataset, 16,583 subgraphs were constructed. Table 3 and Table 4 present the Micro-Prec., Micro-Rec., and Micro-F1 of the compared methods on both datasets. The bolded parts represent the best results. The results show that the proposed Bit-CHetG model outperforms all comparison methods, highlighting the advantages of introducing heterogeneous networks and contrastive learning in the task of Bitcoin money laundering group detection.

As expected, the random forest approach based on primitive features yields the worst results, mainly because it ignores the topology of the transaction network since there are intricate feature interactions. The GNN-based approach improves on this aspect. The results confirm that money laundering group detection using GCN is effective, which emphasizes the reason why we chose GCN as the base encoder for the Bit-CHetG model. Inspection-L follows the framework of DGI [66] and is the first algorithm to apply a self-supervised GNN to the Bitcoin money laundering detection problem. SubGNN is a well-known subgraph detection algorithm and is a state-of-the-art benchmark to validate the superiority of the proposed Bit-CHetG model. It reasons about the topological properties of a given subgraph but lacks a specially designed subgraph sampling algorithm. Tsgn focuses on the subgraph pattern recognition of cryptocurrencies, but the designed Transaction SubGraph Network only collects the 1-hop neighborhood information, which works well in cryptocurrency phishing account identification. However, when applied to cryptocurrency money laundering detection, it is not as effective as our Bit-CHetG algorithm. Based on the results of Table 3, the proposed Bit-CHetG outperforms all the comparison methods, improving the Micro-F1 on the Elliptic dataset by 12%, 7%, 6%, and 5% compared to GCN, Inspection-L, SubGNN, and Tsgn, respectively.

The BlockSec dataset employs two commonly used heterogeneous network models as comparison algorithms, i.e., HAN and MAGNN. In addition to this, by fusing address and transaction features, we apply the homogeneous subgraph algorithms SubGNN and Tsgn to the BlockSec dataset as well. Compared with the basic GCN method, HAN and MAGNN have improved the performance on the BlockSec dataset considering the heterogeneity of the address–transaction graph. In particular, the performance improvement is more significant for MAGNN by employing self-supervised tasks. However, compared to our Bit-CHetG algorithm, HAN and MAGNN do not design specialized metapaths for our Bitcoin money laundering group detection task. As the result in Table 4, the Micro-F1 of Bit-CHetG proposed in this paper improves 10% and 7% compared to HAN as well as MAGNN, and 9% and 7% compared to SubGNN as well as Tsgn, respectively.

Meanwhile, the bottom of Table 3 and Table 4 shows additional experiments to verify the performance of Bit-CHetG, where Bit-CHetG (Reg + Cl + Aug) is the complete algorithm including regression loss, contrastive loss, and graph augmentation, Bit-CHetG (Reg + Cl) removes data augmentation, and Bit-CHetG (Reg) removes data augmentation and contrastive loss. Bit-CHetG (polytree) replaces tree structure sampling with polytree sampling, which may contain multiple parent nodes. The Bit-CHetG model proposed in this paper performs well in both homogeneous graphs containing only transaction information and heterogeneous graphs containing both transaction and address information. The advantages of Bit-CHetG can be attributed to four main reasons:Bit-CHetG selects the appropriate subgraph sampling structure. As shown in Section 5.2, we have chosen the tree structure as the detection unit. The results of Bit-CHetG (polytree) in Table 3 and Table 4 show that the polytree structure as subgraph is inferior to the tree structure. This is because the polytree contains more interaction information which leads to interference in recognizing illicit and licit patterns.Bit-CHetG introduces a contrastive loss in addition to the original regression loss. As shown in the results of Bit-CHetG (Reg) as well as Bit-CHetG (Reg + Cl + Aug) in Table 3 and Table 4, contrastive learning and graph augmentation help the model better learn the commonalities between the same samples and the differences between different samples and thus generates high-quality feature representations.Bit-CHetG employs a flexible data augmentation strategy. By randomly adding or removing edges, we can simulate erroneous transactions or transactions deliberately interfered with by money launderers to conceal their activities. By simulating the noise during transactions through data augmentation, the results of Bit-CHetG (Reg + Cl + Aug) in Table 3 and Table 4 are more robust than those of Bit-CHetG (Reg + Cl).Bit-CHetG purposefully designed Metapaths. For UTXO, the smallest trading unit of Bitcoin, we design ATA-Metapath and TAT-Metapath to directly detect money laundering groups. Compared with the above comparison algorithms that acquire node characterization before applying it to downstream tasks, our approach significantly improves the effectiveness.

### 5.4. Ablation Study

In this part, we first validate the effectiveness of the AFA module and the CL module, and then conduct experiments to select the optimal parameters, one is the optimal number of layers of the graph neural network, and the other is the optimal sample size of the subgraph.

To validate the effectiveness of various modules within Bit-CHetG, we conducted comparisons with the following variants:

**Bit-CHetG_NA**: This variant only employs the transaction record information and disregards the account address information. The AFA component is subjected to masking, resulting in the degradation of the heterogeneous graph to a homogeneous graph.

**Bit-CHetG_NC**: In this variant, the contrastive learning module is removed, rendering the model unable to capture distinctions between instances of different classes and diminishing the quality of representations.

Experimental results are presented in Table 5. The observations are as follows:The introduction of contrastive learning in the Bit-CHetG model yields significant improvements over Bit-CHetG_NC. Specifically, Micro-Prec., Micro-Rec., and Micro-F1 increased by 5.8%, 2.5%, and 4.3%, respectively, highlighting the beneficial impact of contrastive learning.In comparison to Bit-CHetG, the Micro-Prec., Micro-Rec., and Micro-F1 of Bit-CHetG_NA decreased by 3.2%, 1.4%, and 2.3%, respectively. When contrastive learning is directly applied to the transaction graph without auxiliary account information, the model achieves only moderate predictive accuracy.

**Table 5 entropy-26-00211-t005:** Result of the ablation study.

Methods	Micro-Prec.	Micro-Rec.	Micro-F1
Bit-CHetG	**0.825** ^1^	**0.760** ^1^	**0.815** ^1^
Bit-CHetG_NC	0.767	0.735	0.772
Bit-CHetG_NA	0.793	0.746	0.792

^1^ Number in bold represent optimal performance.

These results highlight the importance of introducing different modules into the Bit-CHetG model to improve the prediction performance. Meanwhile, Figure 8 shows the confusion matrix about predicted and true labels from the baseline GNN model and the Bit-CHetG model in three different settings. Bit-CHetG shows more concentration along the main diagonal compared to Bit-CHetG_NA and Bit-CHetG_NC. This concentration indicates higher accuracy. In the money laundering group detection task, more attention should be paid to the detection of subgraphs with high illicit rates. It is worth noting that our Bit-CHetG has a darker heatmap color in the lower right corner of the confusion matrix compared to the simple GNN model, which indicates a more accurate identification of subgraphs with high illicit rates.

Next, we performed some optimal parameter experiments. Firstly, The effect of different GCN layers on the Micro-Prec. was first evaluated. Table 6 summarizes the accuracy results on the BlockSec dataset, where $k_{1}$ denotes the number of GCN layers in the TSE component and $k_{2}$ denotes the number of GCN layers in the AFA component. Thus, the optimal setting for displaying the number of GCN layers for both components is 2.

Furthermore, we consider the optimal size of the transaction subgraph for sampling. Figure 9 depicts the variation of weighted accuracy with the size of the transaction subgraph (*N*) for both the BlockSec dataset and the Elliptic dataset. It can be seen that when *N* is set to 5 for the BlockSec dataset and 10 for the Elliptic dataset, both precision and stability are at their best.

## 6. Discussion

As of 9 August 2022, the size of the entire transaction record of Bitcoin, i.e., the blockchain, is 420 GB, with an average growth rate of 129% [29]. The rule-based manual money laundering detection methods used in the industry are time-consuming and resource-intensive, and there is an urgent need for more efficient methods to detect Bitcoin money laundering. Our algorithm aims to predict the illicit rate of subgraphs, significantly reducing computational costs. Our algorithm will play a crucial role in screening the vast data flow of cryptocurrencies, and subgraphs predicted to have higher illicit rates will be added to a watchlist.

Our proposed algorithm performs exceptionally well in experiments. As illustrated in Section 5.2, the patterns discovered can effectively distinguish money laundering groups from non-money laundering groups. The transparency of the blockchain and the generalization ability of neural networks ensure the practical application of our algorithm. Firstly, cryptocurrency transaction information is recorded in real-time on the blockchain, with public transparency. We can obtain the initiating wallet address ID, receiving wallet address ID, and transaction record ID for a transaction on public platforms (e.g., Blockchain.com [67]). This forms the basis for further mining the original features of IDs and constructing the address–transaction heterogeneous graph, as detailed in Section 3. Secondly, in the paper, we utilized two labeled datasets to train the proposed deep learning network model, Bit-CHetG. The extensive time span covered by these datasets ensures that the model learns optimal parameters with high generalization ability. In practical applications, when dealing with unlabeled transaction graphs, after the subgraphs are sampled, we can directly calculate the illicit rate of the transaction subgraph using those retained model parameters.

The money laundering group detection paradigm based on subgraphs, once applied in reality, will significantly reduce the complexity of cryptocurrency AML. This regulation of cryptocurrency money laundering will further maintain the stability of the global financial system, resist criminal activities, and enhance international security.

## 7. Conclusions

In this paper, we mine the pattern of money laundering and introduce a feasible subgraph sampling approach. We found that a tree-structured sampling approach can distinguish the typical patterns of money laundering groups. Based on this, we propose a novel model to detect Bitcoin money laundering groups, named Bit-CHetG. The model is a subgraph-based graph neural network approach that combines heterogeneous graphs as well as contrastive learning. Experimental results show that our algorithm is very effective and robust in detecting illicit groups in two datasets, significantly outperforming other algorithms.

However, our graph neural network algorithm does not adequately consider the direction of edges in the transaction graph, which is a crucial detail since it provides essential information about the flow of currency. To further enhance the performance of our model, we plan to develop a directed network model in the future, aiming to more accurately capture the directional features of transactions. Furthermore, the topology of the Bitcoin transaction network evolves. This dynamism represents another crucial aspect of our research. In future studies, we intend to incorporate key temporal information to construct a dynamic graph neural network model. This will provide a more comprehensive perspective for our analyses and further improve the performance of our model. 

## Figures and Tables

**Figure 3 entropy-26-00211-f003:**
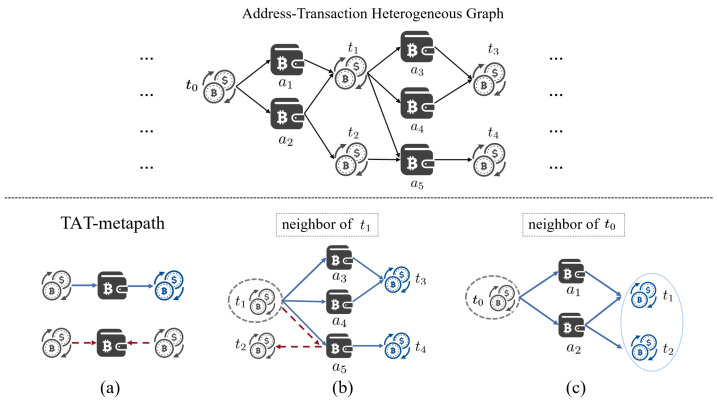
Expansion of the address–transaction heterogeneous graph: (**a**) the TAT-metapath, disregarding the $T\overset{E_{T\to A}}{⟶}A\overset{E_{T\to A}}{⟵}T$ represented by the red dashed line; (**b**) the neighbor of $t_{1}$, where $t_{2}$ is the neglected same-level neighbor; and (**c**) the neighbor of $t_{0}$, where $t_{1}$ and $t_{2}$ is the downstream neighbor framed by a blue ellipse.

**Figure 4 entropy-26-00211-f004:**
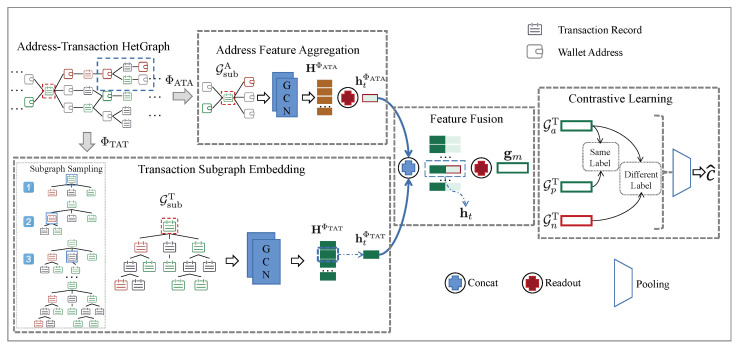
Bit-CHetG. Including four components: transaction subgraph embedding, address feature aggregation, feature fusion, and contrastive learning. The blue box in the address–transaction HetGraph illustrates that one wallet may connect to multiple transactions, and the red dotted line frames the target transaction node.

**Figure 5 entropy-26-00211-f005:**
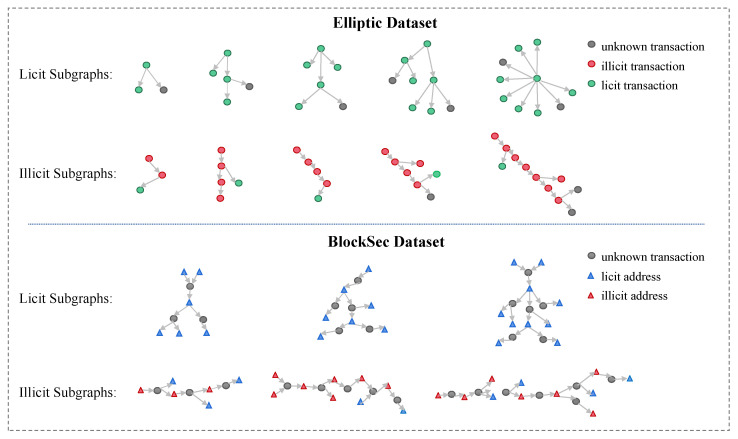
Typical examples of subgraphs. Examples of different topologies of licit and illicit subgraphs in two datasets. The number of sampling hops *n* is increasing from left to right.

**Figure 6 entropy-26-00211-f006:**
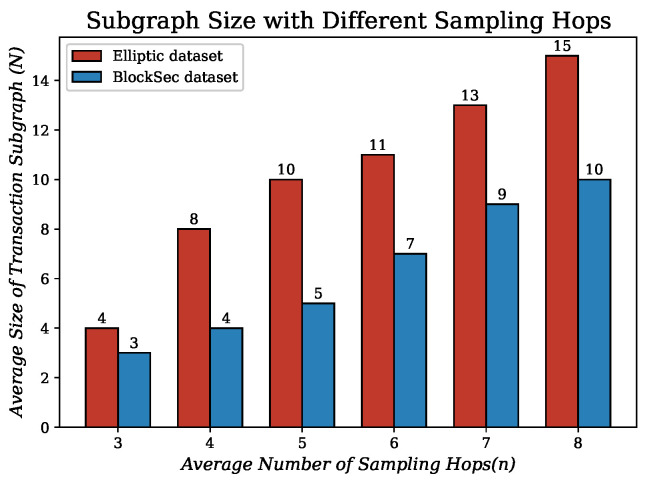
Relationship between the size of transaction subgraph (*N*) and the number of hops (*n*) when sampling a transaction subgraph.

**Figure 7 entropy-26-00211-f007:**
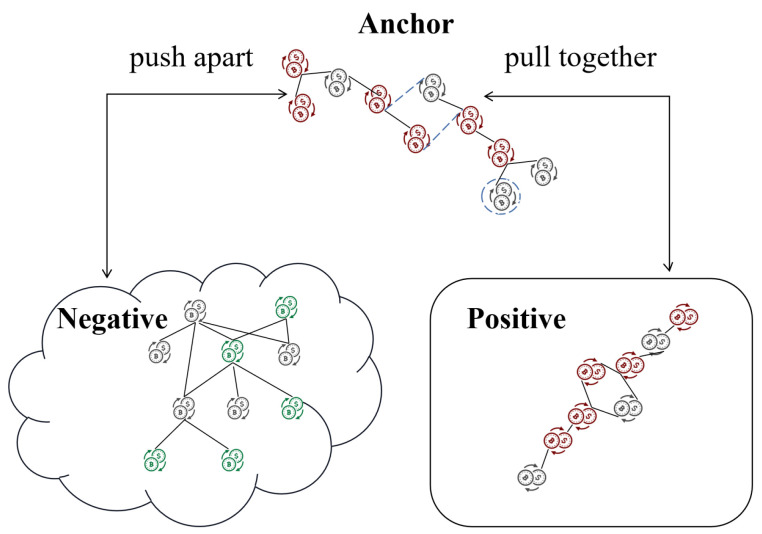
Supervised graph contrastive learning. Pull together the positive sample (same label) and push apart the negative sample (different label).

**Figure 8 entropy-26-00211-f008:**
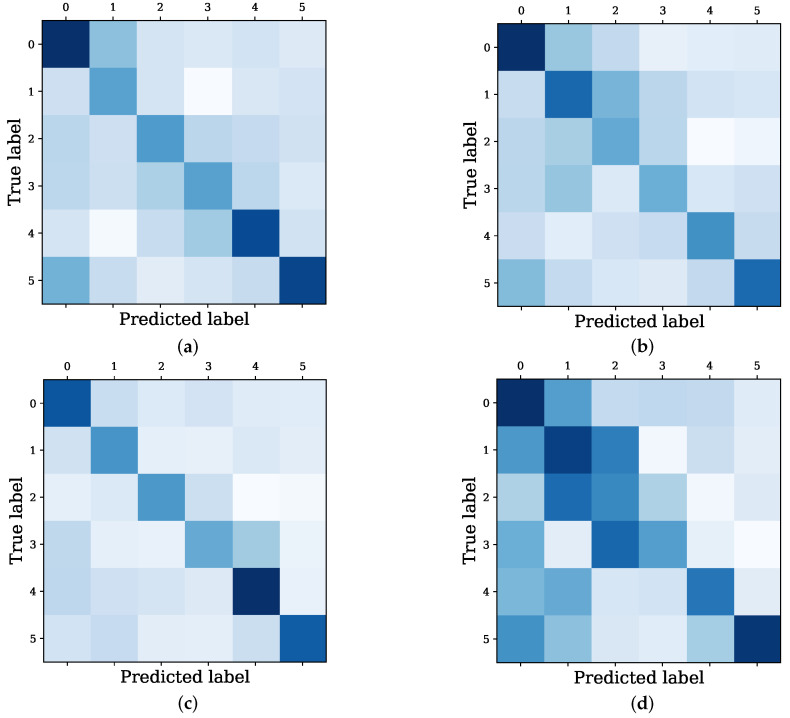
Confusion matrix of the predicted label and true label from the Bit-CHetG model in three different settings and GNN: (**a**) Bit-CHetG_NA; (**b**) Bit-CHetG_NC; (**c**) Bit-CHetG; and (**d**) GNN. The darker the color, the higher the correlation between the predicted label and the real label.

**Figure 9 entropy-26-00211-f009:**
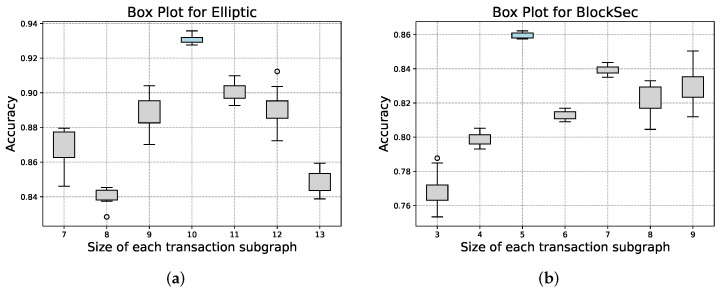
Boxplot of the size of the transaction subgraph vs. accuracy: (**a**) Elliptic dataset; (**b**) BlockSec dataset.

**Table 1 entropy-26-00211-t001:** Notation and definition.

Notation	Definition
$G_{\mathrm{het}}=\left\{T,A,E_{T\to A},E_{A\to T}\right\}$	Address–transaction heterogeneous graph
$G_{\mathrm{sub}}^{T}=\left\{G_{1}^{T},\ldots ,G_{m}^{T},\ldots ,G_{M}^{T}\right\}$	Set of transaction subgraphs
$G_{\mathrm{sub}}^{A}=\left\{G_{1}^{A},\ldots ,G_{t}^{A},\ldots ,G_{\left|T\right|}^{A}\right\}$	Set of associated address subgraphs
$\Phi_{\mathrm{ATA}}$	Two types of metapaths: $A\overset{E_{A\to T}}{⟶}T\overset{E_{T\to A}}{⟶}A$ or $A\overset{E_{A\to T}}{⟶}T\overset{E_{A\to T}}{⟵}A$
$\Phi_{\mathrm{TAT}}$	One type of metapath: $T\overset{E_{T\to A}}{⟶}A\overset{E_{A\to T}}{⟶}T$
$H^{\Phi_{\mathrm{TAT}}}$	The embedding matrix of transaction subgraph under $\Phi_{\mathrm{TAT}}$
$h_{t}^{\Phi_{\mathrm{TAT}}}$	The topological embedding representation of transaction node *t* under $\Phi_{\mathrm{TAT}}$
$H^{\Phi_{\mathrm{ATA}}}$	The embedding matrix of associated address subgraph under $\Phi_{\mathrm{ATA}}$
$h_{t}^{\Phi_{\mathrm{ATA}}}$	The associated address representation of the central transaction node *t* under $\Phi_{\mathrm{ATA}}$
$h_{t}$	The node-level fusion feature vector for transaction node *t*
$g_{m}$	The graph-level fusion feature vector for the *m*-th transaction subgraph

**Table 2 entropy-26-00211-t002:** Statistical information of the datasets.

Dataset	Number of Transactions	Number of Addresses	Number of Subgraphs	Size of Transaction Subgraph
Elliptic	203,769	None	24,533	10
BlockSec	16,674,890	100,061	16,583	5

**Table 3 entropy-26-00211-t003:** Result of Elliptic dataset.

Methods	Micro-Prec.	Micro-Rec.	Micro-F1
Random forest	0.803	0.711	0.694
GCN	0.812	0.801	0.799
Inspection-L	0.869	0.836	0.851
SubGNN	0.865	0.843	0.858
Tsgn	0.879	0.854	0.867
Bit-CHetG (Reg + Cl + Aug)	0.905	**0.893** ^1^	**0.919** ^1^
Bit-CHetG (Reg + Cl)	**0.914** ^1^	0.872	0.889
Bit-CHetG (Reg)	0.873	0.851	0.869
Bit-CHetG (polytree)	0.871	0.841	0.858

^1^ Number in bold represent optimal performance.

**Table 4 entropy-26-00211-t004:** Result of BlockSec dataset.

Methods	Micro-Prec.	Micro-Rec.	Micro-F1
GCN	0.701	0.696	0.699
SubGNN	0.742	0.712	0.722
Tsgn	0.749	0.723	0.741
HAN	0.742	0.712	0.718
MAGNN	0.751	0.736	0.745
Bit-CHetG (Reg + Cl + Aug)	**0.825** ^1^	0.760	**0.815** ^1^
Bit-CHetG (Reg + Cl)	0.807	**0.772** ^1^	0.802
Bit-CHetG (Reg)	0.791	0.751	0.789
Bit-CHetG (polytree)	0.771	0.740	0.758

^1^ Number in bold represent optimal performance.

**Table 6 entropy-26-00211-t006:** Accuracy at different number of GCN layers.

	*k* _2_	1	2	3
*k* _1_				
1		0.81	0.84	0.82
2		0.82	**0.86** ^1^	0.84
3		0.81	0.83	0.85

^1^ Number in bold represent optimal performance.

## Data Availability

Data are contained within the article. For the code and dataset please refer to the link of github: https://github.com/oy2020/Bit-CHetG.

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
