# Peer review of "Bitcoin Money Laundering Detection via Subgraph Contrastive Learning"

_entropy, 2024, doi:10.3390/e26030211_

Round 1
Reviewer 1 Report
Comments and Suggestions for Authors
The structure of the paper is good and the research interest is good in timely manner. There are few points that I suggest to improve:
1. Introduction should be more focused on why the paper should be conducted. The current introduction is more likely algorithm based approach. However, there should be social-context as well as the technical-context of the issue. The technical-context background is placed within 3.Problem. Therefore, I suggested to explore the previous literature about the money laundering and relative algorithms. There are many articles recently published to suggest the money laundering. So, the current paper should talk about the social issue and how the previous papers investigated the issue with algorithms.
2. There should be discussion before the conclusion. As mentioned above, the issue of money laundering is not only for the algorithm or technical issue. It should be conbined with social issue and economic issue as well. Therefore, the current paper should talk how the results are associated with helping the issue socially or economically.
Reviewer 2 Report
Comments and Suggestions for Authors
Focusing on the identification of money laundering groups in bitcoin, this paper proposes a subgraph-based heterogeneous graph comparative learning algorithm that reveals the differences in transaction topology between money laundering groups and normal users. By combining the topological embedding of transaction subgraphs and the associated address representation of transaction nodes, and introducing supervised comparative learning to reduce the effect of noise, the effectiveness of detecting money laundering groups is improved. Finally, the effectiveness of the programme is experimentally demonstrated and the problem chosen is meaningful and interesting. The paper is well structured and well written.
However, there are still some points that need to be improved.
1. Subsection 2.1 should provide a clearer delineation of related work, as well as a detailed explanation of the gaps in the existing literature and the innovations brought by this paper's approach. In addition, more concrete examples of how bitcoin money laundering differs from other forms of cybercrime and how this affects detection systems will strengthen this section.
2. Figure 2 needs more intuitive textual annotation and graphical representation than provided to clearly describe the neighbors of t1 and a1, for example by using different colours to indicate the direction of the lines.
3. In definition 2 of section 3, there should be a deeper reason why the path T->A<-T should be ignored.
4. For Section 4.2, more technical details on address feature aggregation would be helpful.
5. I suggest a more detailed description of the experimental environment and parameter settings, as well as the procedure for working with the two datasets.
6. The study conducted by the authors is experimental in nature. What are the chances of the research presented in the article being of practical use?
Comments on the Quality of English LanguageMinor editing of English language required.
